

# Using a real-world network to model the trade-off between stay-at-home restriction, vaccination, social distancing and working hours on COVID-19 dynamics

Ramin Nashebi[1], Murat Sari[1,2] and Seyfullah Kotil[3]

[1] Department of Mathematics, Yildiz Technical University, Istanbul, Turkey
[2] Department of Mathematics Engineering, Faculty of Science and Letters, Istanbul Technical University, Istanbul, Turkey
[3] Department of Biophysics, School of Medicine, Bahcesehir University, Istanbul, Turkey

Corresponding author
Seyfullah Kotil, enesseyfullah.kotil@med.bau.edu.tr

## ABSTRACT

**Background**. Human behaviour, economic activity, vaccination, and social distancing are inseparably entangled in epidemic management. This study aims to investigate the effects of various parameters such as stay-at-home restrictions, work hours, vaccination, and social distance on the containment of pandemics such as COVID-19.

**Methods**. To achieve this, we have developed an agent based model based on a time-dynamic graph with stochastic transmission events. The graph is constructed from a real-world social network. The edges of graph have been categorized into three categories: home, workplaces, and social environment. The conditions needed to mitigate the spread of wild-type COVID-19 and the delta variant have been analyzed. Our purposeful agent based model has carefully executed tens of thousands of individual-based simulations. We propose simple relationships for the trade-offs between effective reproduction number ($R_e$), transmission rate, working hours, vaccination, and stay-at-home restrictions.

**Results**. We have found that the effect of a 13.6% increase in vaccination for wild-type (WT) COVID-19 is equivalent to reducing four hours of work or a one-day stay-at-home restriction. For the delta, 20.2% vaccination has the same effect. Also, since we can keep track of household and non-household infections, we observed that the change in household transmission rate does not significantly alter the $R_e$. Household infections are not limited by transmission rate due to the high frequency of connections. For the specifications of COVID-19, the $R_e$ depends on the non-household transmissions rate.

**Conclusions**. Our findings highlight that decreasing working hours is the least effective among the non-pharmaceutical interventions. Our results suggest that policymakers decrease work-related activities as a last resort and should probably not do so when the effects are minimal, as shown. Furthermore, the enforcement of stay-at-home restrictions is moderately effective and can be used in conjunction with other measures if absolutely necessary.

## INTRODUCTION

Human behavior in households, workplaces, social environment during weekends and weekdays have a vital role in the dissemination of infectious diseases such as Middle East Respiratory Syndrome (MERS) (*Liu et al., 2020*), H1N1 influenza (*Dotis & Roilides, 2009*), severe acute respiratory syndrome (SARS) (*NHS, 2019*), and the current acute respiratory syndrome coronavirus 2 (SARS-CoV-2) (*Liu et al., 2020*). Household members have frequent and intimate contact, making the disease spread rapidly within households (*Nele et al., 2018*; *Seventer, Maguire & Hochberg, 2017*). Developing strategies for preventing infectious diseases is a priority of health organizations. There are three general containment strategies for avoiding and mitigating contagious diseases: antiviral, vaccine, and non-pharmaceutical measures (*Erguson et al., 2006*). The non-pharmaceutical measures include a wide range of policies such as changing the number of work hours, limiting transmissibility between individuals by distancing measures, and stay at home (short-term) restrictions (*Nande et al., 2021*; *Guest, Del Rio & Sanchez, 2020*; *Iavicoli et al., 2021*).

A stay-at-home policy is frequently used (*Guest, Del Rio & Sanchez, 2020*; *Tognotti, 2013*; *Markel et al., 2007*; *Savaris et al., 2021*). The World Health Organization (*WHO, 2020*; *World Health Organization, 2020*) and many local authorities (*Governor's Press Office, 2020*; *Ministry of Housing, Communities & Local Government, 2020*) supported and encouraged stay-at-home measures. Governmental policies upon stay-at-home orders are grouped into four (*Our World In Data, 2022*) categories: no measures implemented concerning staying at home, recommended to stay at home, moderate restriction concerning staying at home (people can do their daily exercise, grocery shopping, and 'essential' trips), and high restriction regarding staying at home (people allowed to leave only once every few days, or only one person can go at a time). The same countries modified the lengthy lockdown to a short-term stay at home orders, such as France (*The Connexion, 2022*; *France 24, 2021*), Turkey (*Daily Sabah, 2021*; *United States of America Department of State, 2021*), and India (*Nayak, 2021*). These countries recommend stay-at-home orders only during weekends to stem coronavirus cases. For example, the first weekend stay-at-home restriction, in Turkey, took place between April 10, 2020 at 24:00 and April 12, 2020 at 24:00 (*UNHCR, 2020*). On the downside, stay-at-home orders, limiting work hours and distancing measures affect mental health negatively, physical health, and the economy (*Alwan et al., 2020*). Understanding the interplay of the mentioned non-pharmaceutical policies and vaccination is essential in the containment of the disease and successfully managing the economy. Since the experimental study is not compliant to investigate the social dynamic, control, and management of disease among humans, mathematical models are crucial to quantify and investigate such effects (*Funk, Salathé & Jansen, 2010*).

The role of stay-at-home (short-term) orders, limiting work hours, distancing measures, and vaccination in disseminating COVID-19 requires modelling the interplay between agents and their environment. The social and spatial heterogeneity of agents and the interaction between agents. These factors make it a complex phenomenon (*Venkatramanan*

et al., 2018). The problem can be represented by agent based models simulated on a time-resolved contact network with stochastic events. We have aimed for a model where the duration of stay-at-home restrictions, number of working hours, distance measures, and vaccination can be varied independently on an individual level.

Agent based modelling is a computational approach to modelling complex systems with autonomous agent interactions (Macal & North, 2010; Tracy, Cerdá & Keyes, 2018). Consequently, agent based models are essential tools for understanding the impact of human behavior in transmitting infectious diseases in different environments such as households, workplaces, and social environments. Aleta et al. (2020) used an agent based model with three layers: school, workplace, and household. They used their model to investigate the influence of the closure of schools and stay-at-home restrictions. Another study by Hoertel et al. (2020) developed a stochastic agent based model and ran it over a synthetical social network. Braun et al. (2020) developed a network-based, agent based model. Using this model, they simulate the synthetic Watts–Strogatz small-world network to catch the efficiency of social distancing, personal protective equipment, and quarantining. Some mathematical models and meta-analyses analyzed the dynamic of COVID-19 inside and outside of households (Shen et al., 2020; Nande et al., 2021; Grijalva et al., 2020; Bulfone et al., 2020a; Qian et al., 2021; Farthing & Lanzas, 2021). Many studies found that the probability of indoor transmission was remarkably high compared to outdoors (Aleta et al., 2020).

Basic reproduction number ($R_o$) indicates infectious diseases' contagiousness or transmissibility when the population is only susceptible (Delamater et al., 2019). At the same time, the effective reproduction number ($R_e$) estimates an epidemic's growth rate, which is influenced by the containment strategies, herd immunity and any other factor (Delamater et al., 2019). Thus, estimates of COVID-19 are not exclusively determined by the pathogen, and variability depends on local socio-behavioral and environmental settings (Sy, White & Nichols, 2021). Anderson RM (Anderson & May, 1992) and May RM (Anderson & May, 1992) calculate the effective reproduction number as a function of contacts, transmission rate, and transmission duration. We used it as a measurement parameter to explore the role of stay-at-home restriction, working hours, social distancing, and vaccination in the containment of COVID-19.

There have been many published papers to study the role of vaccination (Colomer et al., 2021), social distancing (Nande et al., 2021), and household (Nande et al., 2021; Farthing & Lanzas, 2021) in understanding the dynamic and control of the SARS-CoV-2 virus. Several authors have designed and used simple models (Das et al., 2021; Moore et al., 2021b; Betti et al., 2021), complex models (Colomer et al., 2021; Sonabend et al., 2021; Breban, Riou & Fontanet, 2013), and multiscale models (Kou et al., 2021; Kotil, 2021) to simulate the trade-off between pharmaceutical (vaccination) and non-pharmaceutical (social distancing, stay-at-home restriction, decrease in working hours) intervention in the containment of COVID-19 pandemic. This study presents an agent based model based on a time-dynamic network with stochastic transmission events to analyze the interplay between pharmaceutical and non-pharmaceutical interventions. We made thousands of carefully executed individual level simulations of multiscale modelling on the real network. The

benefit of the modelling is that the following factors are adjustable: stay-at-home restriction, working hours, vaccination, and social distancing. The simulations that were made depend on four parameters: Decrease in working hours (DW), social distancing measure (SDM), stay-at-home restriction (SH), and vaccination ratio (Vac). The multidimensionality of the problem hinders clear understanding. We represent all our results that show the trade-offs between complex phenomena in a simple mathematical expression. This simple function links the modelled forces with ($R_e$) from the generated data. Additionally, since we can distinguish between household and non-household infections, some exciting observations include that the household infections provide resilience for epidemic eradication but do not contribute significantly in spreading.

## MODEL AND METHODS

### Real-world social network data description and classification

We use the BBC documentary, Contagion (*Soderberg, 2011*), to make our simulation more realistic of human interaction patterns. The *BBC (2017)* demonstrates the social network data of 469 volunteers from Haslemere, England (*Klepac, Kissler & Gog, 2018*). The data is not categorized as where the interactions occur. Our main goal is to order each interaction, whether it occurs in workplaces, social environments, or households. By classifying the edges of the graph, we can understand which of the settings and behaviors of staying in that environment would affect the COVID-19 dynamics.

The Haslemere dataset consists of the pairwise distances of up to 50m resolution between 469 volunteers from Haslemere, England, at five-minute intervals over three consecutive days (Thursday 12 Oct–Saturday 14 Oct 2017). It gives users' data for sixteen daytime hours only, between 07:00:00am and 22:55:00pm, excluding the hours between 11 pm and 7 am. There are 576-time points for each user. According to the 2011 UK census, volunteers of the Haslemere dataset established a sample of 4.2% of the total population of Haslemere. Participants downloaded the BBC Pandemic mobile phone app and then went about their daily business with the app running in the background. The study was restricted to volunteers of at least 16 or 13 ages having their parental consent. The pairwise distances between volunteers were calculated using the Haversine formula for great-circle geographic distance and are based on the most accurate GPS coordinates that the volunteers' mobile phones could provide (*Kissler et al., 2019*; *Kissler et al., 2018*).

We want to categorize this network into households, workplaces, and social environments. In other words, we classify each edge and contact into three categories. We choose contacts with an average pairwise distance of 20m or less, making 3245 unique contacts. Our primary analysis identifies contacts whose average pairwise distances are 5m or less because we aimed to capture household encounters and understand their daily routine and behavior. To investigate the behavior of everyone during these three days, we developed a visualization method (see Fig. S2). After investigation, we developed procedures (see Table S1) to categorize the network edges and contacts in the household, workplace, and social environment. There are 1,350 unique contacts, which occurred between 1m and 5m. We use the visualization method to classify 123, 514, and 713 of them as household, workplace, and social environment contacts, respectively.

We established an automated classification algorithm to classify the remaining 1,895 unique encounters, which occurred between 6m and 20m distances. We developed the algorithm using procedures from the visualization method, see Fig. S1 for algorithm pseudocode. From 1,895 unique contacts, 52, 790, 1,003 of them occurred in the household, workplaces, and social environments. We create a confusion matrix to evaluate the algorithm's performance by choosing the classified data points (using the visualization method) as actual data and the algorithm classified data points as predicted data. Overall, 91.73% of the predictions of our algorithm are correct, and 8.27% are incorrect, according to the confusion matrix of the algorithm concerning data points classified by the visualization method (see Fig. S2). The distribution of household, workplace and social environment interactions after classification is illustrated in Fig. 1B.

## Agent based model

We have developed a discrete-time stochastic agent based model, parameterized to simulate distinct types of COVID-19 outbreaks across the Haslemere data set. An agent in our simulation can be in the following states: $E(t)$ (exposed), $PS(t)$ (pre-symptomatic, documented), $A(t)$ (asymptomatic, undocumented), $S(t)$ (symptomatic, documented) $H(t)$ (hospitalized) and $R(t)$ (recovered) (Fig. 1C). In our model, symptomatic individuals are those who show symptoms; those who are not considered undocumented (asymptomatic). The agent based model starts with an exposed individual. Initially, the $j$ th individual is exposed to the virus, and he/she cannot infect others during his/her latent period for $d_1 = 2.7$ days (see Table S2). After the latency period finishes, the $j$ th individual ends up in one of the two branches: pre-symptomatic with a ratio of s or asymptomatic with $(1 - s)$. When the period of delay from the onset finishes, we generate a random number $\varepsilon$, and there are two probabilities for the $j$ th individual to proceed; the first one, if $\varepsilon < 1$-s, he/she proceeds to the asymptomatic stage. The infectious period of the asymptomatic stage is $d_3 = 5.4$ days (see in Table S2). Through the infectious period of the asymptomatic stage, when the $j$ th individual comes in contact with the $i$ th individual, then we infect the $i$ th individual with probability $\mu_H Ph_{j,i}$ if the edge is classified as household and $\mu_o Po_{j,i}$ if non-household. The $\mu_H$ and $\mu_o$ are the reduction factor for asymptomatic transmission in households and non-household. $Ph_{j,i}$ and $Po_{j,i}$ are the probability that $j$ th individual infects $i$ th individual of his/her contacts in household and non-household. When the infectious period of the asymptomatic stage finishes, the $j$ th individual proceeds to the recovered stage.

Alternatively, the $j$ th individual can be in the pre-symptomatic stage. The infectious period of the pre-symptomatic stage is $d_2 = 2.4$ (Ding et al., 2021). The infection probabilities are $Ph_{j,i}$ and $Po_{j,i}$ in household and non-household. When the pre-symptomatic stage finishes, the $j$ th individual proceeds to the symptomatic stage and stays for $d_4 = 3$ days (Faes et al., 2020). The infection probabilities are the same as in the pre-symptomatic stage. When the delay period from symptomatic finishes, $j$ th individual proceeds to the hospitalization stage where he/she will recover or die. The hospital stage does not necessarily mean that the agent is hospitalized. All reported cases end up in the

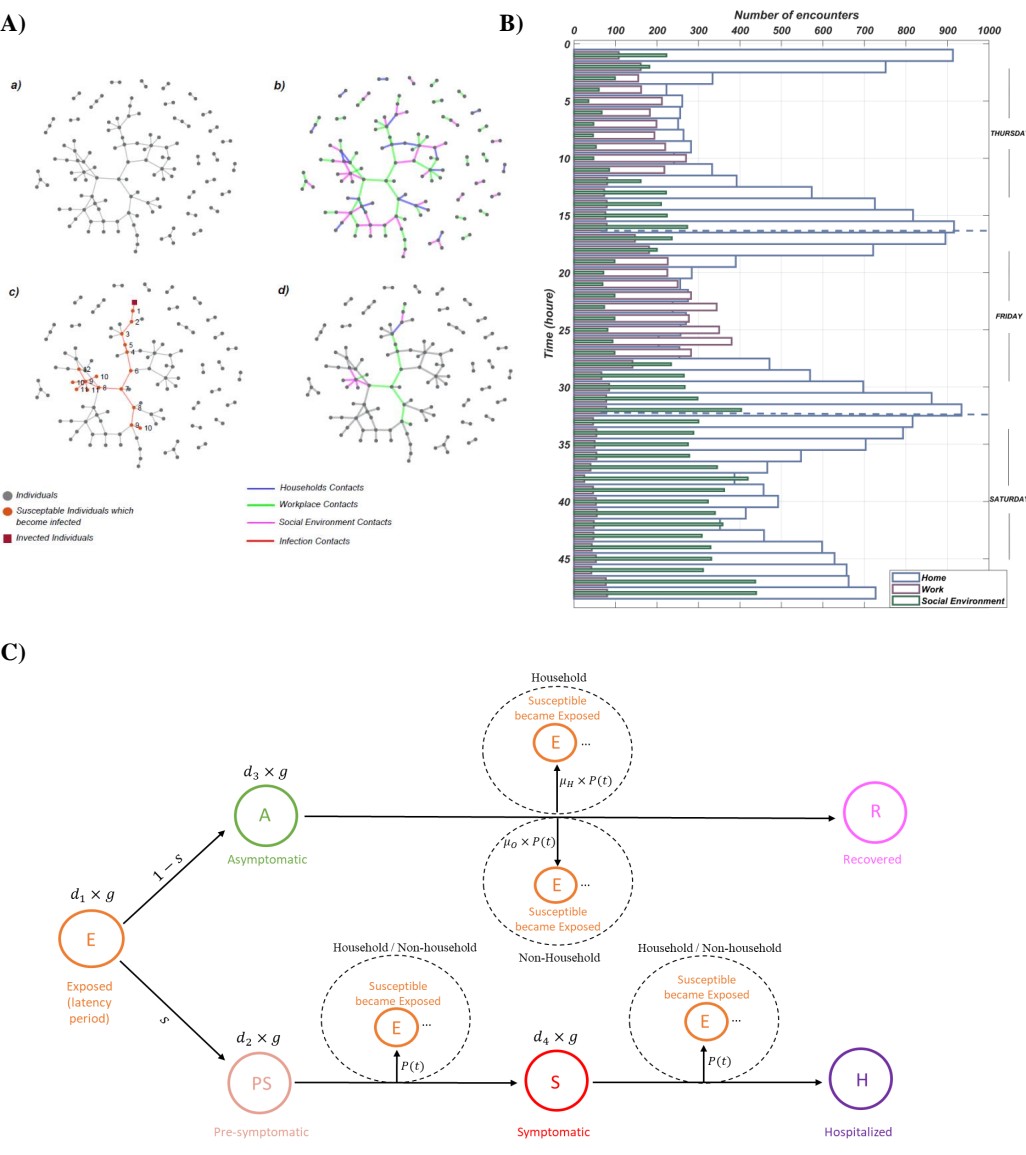

Figure 1 **Methodology of our work.** (Aa) Firstly, we chose and analysed a real-world social network dataset of Haslemere, England. (Ab) Secondly, we classified social network edges into the household (blue edges), workplace (green ages), and social environment (magenta edges). (Ac) After that, we start the simulation with our agent-based model. Here, the dark red square shows exposed individuals, red circles illustrate susceptible individuals infected by interacting with exposed individuals, and light red edges represent infection contacts. We make chains of infection ancestors and descendants from discovered infected agents. Here, two chains of infection ancestor and descendants with 10 cases shown (Ad) Finally, we identify infection occurrence environments. (B) Number of encounters that occurred for three days (Tuesday, Friday, Saturday) after classification of data (C) Graphical representation of our discrete-time, stochastic agent-based model, which starts with an exposed agent. (continued on next page...)
**Figure 1 (…continued)**
After the latency period ($d_1$) finishes, he/she tends to one of the two branches: pre-symptomatic with a ratio of $s$ or asymptomatic with a ratio of (1-$s$). The infectious period of the asymptomatic stage is $d_3 = 5.4$ days, in which he/she infect susceptible hosts in household and non-household (workplace, social environment) with a probability of $\mu_H * P(t)$ and $\mu_O * P(t)$, respectively. After that, he/she proceeds to the recovered stage. If he/she proceeds to the pre-symptomatic stage, he/she will stay in this stage for $d_2 = 2.4$ days. During this period, he/she infect susceptible hosts in household and non-household (workplace, social environment) with the probability of $P(t)$. After the pre-symptomatic period ends, the agent changes the stage to the symptomatic stage. The symptomatic period lasts for $d_4 = 3$ days. Throughout the infectious period of the symptomatic stage, he/she infect susceptible hosts in household and non-household (workplace, social environment) with the probability of $P(t)$. When the delay period from symptomatic finishes agent proceeds to the hospitalization stage, where he/she will recover or die. The default parameter values with their sources were presented in Table S2.

hospital stage, regardless of severity (which is not modelled). The default parameter values for the simulations with their sources were presented in Table S2.

## Constructing contact matrices and simulation with real-world social network data

We construct classification matrices for households, $H_{i,j}$, workplaces, $W_{i,j}$, and social environment, $S_{i,j}$, which we classified from Haslemere data. $H_{i,j} = 1$ if the contact of $j$ th and $i$ th agent occurs in a household environment, and $H_{i,j} = 0$ otherwise. The simulations are made for 5 min time slot. Each contact is checked at each time step if one of the agents is infectious; also, if the other is susceptible, then it is infected by the precomputed probability. There are three days in the Haslemere data: Thursday, Friday, and Saturday. Our simulations take 14 days, so we construct the more prolonged contact network using real data. The weekday contacts are taken from Thursday to Friday. We repeat each day sequentially and in whole. We do not mix the data that is in a single day. The Sunday contacts are replicated from Saturday contacts of the Haslemere data. We start the simulation with two exposed individuals. The first 14 days are designed as a warmup run. After 14 days, we randomly chose two individuals whose states are exposed, presymptomatic, asymptomatic, or symptomatic. Furthermore, the repetitive use of the Haslemere data can be problematic. However, we simulate for a very short period and for small outbreaks. The goal is for our simulations to be independent of one another.

We start with a more realistic initial sample of states by this method. Then, we restart the simulation for another fourteen days. We calculate the effective reproduction number ($R_e$), infection occurrence ratio, and secondary attack rate (SAR) of households from the second set of 14-day simulations. We run five hundred trial simulations for each single scenario of our research (see Table S3). The default parameter values for the simulations were present in Table S2. By starting with only two infected individuals, we aimed that the total cases at the end of 14 days do not exceed 10% of the total population. If the susceptible population size is not high enough, it will cause the reproduction number to be overestimated (*Delamater et al., 2019*). Therefore, we chose a small portion of the population to be infected. In addition, the number of infections should not be too low for reproduction number predictability to be unaffected by stochasticity. For independent events the standard deviation is the root square of the mean. Therefore, although the
maximum number that we decided to infect is a small percentage of the population, its square root should be smaller than its mean. Hence, we set the maximum infection occurrence percentage at 10% of the population, corresponding to 40 people. In line with our decision, even after two years, COVID-19 cases in countries did not reach a substantial percentage (*Mathieu et al., 2020*; *World Health Organization, 2022*).

## Calculating infection probability (*P*) and basic reproduction number (*Ro*)

We specify the probability that a susceptible agent *i* th becomes infected by a *j* th infectious agent in a 5-minute interval is given as a function of distance in Eq. (1) for household and in Eq. (2) for non-household (workplace, social environment).

$$Ph_{j,i}(t) = \begin{cases} \beta_h e^{-(\alpha dis_{j,i})} & dis_{j,i} \leq \theta \\ 0 & dis_{j,i} > \theta \end{cases} \tag{1}$$

$$Po_{j,i}(t) = \begin{cases} \beta_o e^{-(\alpha dis_{j,i})} & dis_{j,i} \leq \theta \\ 0 & dis_{j,i} > \theta \end{cases} . \tag{2}$$

$Ph_{j,i}(t)$ is the probability of infection if the contact occurs in a household. $Po_{j,i}(t)$ is the probability of infection if the contact occurs in the workplace or social environment. Also, $dis_{j,i}$ is the distance in meters between individuals *i* and *j* at time *t*; $\alpha$ , $\beta_h$, $\beta_o$ are distance scaling parameters, transmission probability per 5 min in and out of households, respectively. $\theta$ defines the cutoff distance, after which the infection probability is assumed to be zero. Our social network consists of a pairwise distance of up to 50m. Since we choose only those interactions which occurred less than or equal to 20 m, so $\theta = 20$. Infection probability ranges between 0 and 1. Our network consists of 102,831 interactions in 5 min time intervals for three consecutive days (Tuesday, Friday, and Saturday). We calculate the infection probability for each interaction separately. Effective reproduction number ($R_e$) is estimated directly by counting the descendants of a discovered agent after the simulation is finished, then averaged for 14 days. Percentage reduction of transmission probability is computed as a ratio of the effective-outside transmission probability to WT (default, estimated) transmission rate. Thus

$$D = \frac{\beta_o}{\beta_{WT}} 100. \tag{3}$$

## Estimating infection occurrence ratio

Throughout the simulations, we track down antecedents and descendants of infected agents. Each infection pair is an edge of the classification matrices, $H_{i,j}$, $W_{i,j}$, $S_{i,j}$. Then each classified edge is calculated.

## Calculating household secondary attack rate (SAR)

The household secondary attack rate is defined as the probability of transmission per susceptible household member when a single infected individual is in the house (*Nande*

*et al., 2021*; *Moreland et al., 2020*). The SAR of households calculates the number of cases among contacts of primary cases divided by the total number of primary cases. Our model starts the simulations with ten exposed individuals executed with a daily life contact matrix, but we calculate the SAR of households from the second set of 14-day simulations which start with 2 infected individuals. After that, we tract the infection occurrence environments (household, workplace, social environment). We identify those who are members of different households. We accept those initially exposed individuals who are members of different households as our primary case in each household. At the end of the simulation, we calculate the SAR of each household by the following formula:

$$\frac{\text{Number of cases among contacts of primary cases}}{\text{Total number of contacts of primary cases}} \times 100. \tag{4}$$

## Calculating scaling parameters of distance ($\alpha$) and transmission rate ($\beta$) of COVID-19

We estimate scaling parameters of distance and transmission rate by sampling to obtain a baseline Ro = 2.87 (*Billah, Miah & Khan, 2020*) and $0.46 \leq \text{SAR} \leq 0.72$ (*Kuwelker et al., 2021*), which are the basic reproduction number and secondary attack rate of households of COVID-19, respectively. Figure S4 in supplementary information shows the sampling results for calculating $\alpha$ and $\beta$ constants for COVID-19. To do this, we calculate the absolute error between the estimated model $R_o$ and the exact COVID-19. We also compute the absolute error between the estimated and the exact household SAR model for COVID-19. Finally, we add these two errors and choose $\alpha$ and $\beta$ values that give the smallest error. The estimated values are present in Table S2.

## Changing number of work hours during weekdays

We alter work hours during weekdays by changing the work edges of the network. According to our classification methodology of the network, individuals work for 9 h from 9:00 AM up to 6:00 PM during weekdays. To decrease work hours from 9 h to $9 - i$ ($1 \leq I \leq 9$) hours during weekdays, we identify a home, work, and social environment edges between 06:00 PM and $9 - i$ hour before 6:00 PM of each weekday. Secondly, we randomly change only work edges with work, home, and social environment edges from edges between 6:00 PM and 11:00 PM of each weekday. For example, if we want to decrease work hours from 9 h to 8 h ($i = 1$) during weekdays. Firstly, we identify every workplace, social environment, and household contact that occurred between 06:00 PM hour and 05:00 AM hour. Then we changed only workplace contact with the workplace, social environment and household contacts between 6:00 PM and 11:00 PM.

## Simulating stay-at-home restrictions

We investigate the role of stay-at-home restriction on the epidemic dynamics. Systematically, restricting all days and their combinations is possible. However, we have simulated the most probably scenarios. For example, most countries including Turkey (*Daily Sabah, 2021*; *United States of America Department of State, 2021*), India (*Nayak, 2021*), and France (*The Connexion, 2022*; *France 24, 2021*) have implemented

a stay-at-home restriction only on Saturday and Sunday. We do all that and additionally implement stay-at-home restrictions on Thursday, Friday. Therefore we simulate four stay-at-home restrictions scenarios: restrictions on Sunday, restrictions on Sunday and Saturday, restrictions on Sunday, Saturday and Friday and restrictions on Sunday, Saturday, Thursday, and Friday. To implement stay-at-home restrictions, we replace all non-household (workplace and social environment) contacts that occurred in a 5-minute time step with a household connection that also occurred in that same day.

## Lowering transmission probability, increasing social distancing

The infection probability is parameterized by two parameters, $\beta$ and $\alpha$. The $\alpha$ indicates the probability of decay with the distance of contacts, whereas $\beta$ is the maximum transmission probability (at a distance of 0). We have obtained these parameters by sampling many simulations that fit the COVID-19's Ro and secondary attack rate. We use two $\beta$ parameters, $\beta_h$, and $\beta_o$, to distinguish between the infectiousness in households and outside. In our simulations, we reduce the infectious probability to simulate a population where people reduce the probability of infection by personal social distancing measures. The parameter $\beta$ could be the total virus present near an infectious agent. In contrast, $\alpha$ is the "loss parameter" (due to diffusion and other effects) for distance. Thus, we have chosen to alter the parameter $\beta$ on the total number of virus changes when a person engages in protective measures such as wearing masks. The parameter $\alpha$ is left untouched since the events that lead to the distance loss do not change. Alternative explanations can be made; however, this is how our simulation has been conducted. Whenever we mention a reduction in infectiousness, we reduce the parameter $\beta$.

People do not conduct social distancing in their homes (*Nande et al., 2021*). To simulate this phenomenon, we reduce the total transmission rate by decreasing the non-household transmission ($\beta_o$) while fixing the household transmission rate ($\beta_h$). However, in some simulations, we alter both $\beta_h$, and $\beta_o$ when agents at home are also engaging in personal protection.

## Vaccination and the delta variant

The simulations with delta variants use different values for $\beta$ and $\alpha$ parameters. Again, we have left $\alpha$ untouched. We have changed $\beta$ according to the literature (*CDC, 2021*). It is assumed that the infectious probability of the delta variant is double of the wild type of virus. We have used $\beta$ for the delta variant as 0.33, whereas the wild type was 0.167.

The agents that are vaccinated are chosen randomly. We investigate six vaccination scenarios: no vaccination, 50% of the population vaccinated, 60% of the population vaccinated, 70% of the population vaccinated, 80% of the population vaccinated, and 90% of the population vaccinated. The vaccinated agents have reduced $\beta$ value. We have assumed a 93% and 88% reduction in infectiousness for the wild type and delta variants, respectively (*Lopez Bernal et al., 2021*). So, $\beta$ for the vaccinated, wild type and delta variant is 0.012 and 0.067, respectively, whilst $\alpha$ is unchanged.

## Parameters

**General**: $d_1 = 2.7$ days, $d_2 = 2.4$ days, $d_3 = 5.4$ days, $s = 0.83$, $p = 3$ days, $\alpha = 0.9841$, $\mu_H = 0.696$ $\mu_o = 0.42$, $g = 192$.

Figures 2A–2B: For the first scenario where people implement social distancing measure only in non-household environment, $\beta_h = 0.1672$, $0.001 \leq \beta_o \leq 0.1672$. For the second scenario where population implement social distancing measure in both household and non-household, $0.001 \leq \beta_h \leq 0.1672$, $0.001 \leq \beta_o \leq 0.1672$.

Figure 3: $\beta_h = \beta_o = 0.1672$ for $R_o = 2.87$, $\beta_h = \beta_o = 0.0214$ for $R_0 = 1$, $\beta_h = 0.1672$, $\beta_o = 0.0165$ for $R_0 = 1$, 88% reduction applied to $\beta_o$. Same parameter for random networks.

Figures 5A–5B: $\beta_h = 0.1672$, $0.001 \leq \beta_o \leq 0.1672$.

Figures 7A–7B: $\beta_h = 0.1672$, $0.001 \leq \beta_o \leq 0.1672$.

# RESULTS

This work investigates the impact of stay-at-home restrictions, social distancing, working hours, and vaccination on the effective reproduction number. To do so, we simulate disease dynamics on a real network (Fig. 1A) by using an agent based model (Fig. 1C). The edges of the network and contacts between agents (nodes) are further classified for the place of infections (Fig. 1B). The three categories are household (H), workplace (W), and social environment (S). Firstly, we investigate how three environments contribute to disseminating the disease on an unaltered network. In that analysis, we also vary the social distancing parameter. Secondly, we investigate the effect of stay-at-home orders on weekends. Thirdly, we investigate all parameters together: change of work hours, vaccination, stay-at-home orders, and social distancing.

## Estimating infection occurrence ratio in the household, workplace and social environment

It is crucial to know where infections occur in relation to the effective reproduction number ($R_e$). Here, we count the infection occurring in three environments: households, work, and social.

Our agent based model simulates $R_e$ by independently varying household and non-household transmission rates $\beta_h$ and $\beta_o$, respectively. Squares with a line in Fig. 2A show $R_e$'s dependence on varying $\beta_o$ when $\beta_h$ is constant. Additionally, triangles with a line show the dependence of $R_e$ on varying $\beta_o$ when $\beta_h$ is also changing, $\beta_o = \beta_h$. The latter simulations are made to assess the effect of making the distinction between $\beta_o$ and $\beta_h$. Figure 2A show that, interestingly, until $R_e < 1$, $Re$ does not depend on the decrease of $\beta_h$. Only after $R_e < 1$ further decreasing in $\beta_h$ decreases $R_e$ more than when $\beta_h$ is kept at its maximum.

The household, workplace, and social environment infection occurrence ratio has been counted for $R_e$ (for real networks). Figures 2B and 2C demonstrate that household infections are dominant when $R_e < 1$, and most infections occur in the social environment when $R_e < 2.5$.

We capture the only difference between Fig. 2B ($\beta_h$ kept constant) and Fig. 2C ($\beta_h$ decreases) was in the slight increase of household infection occurrence ratio when $\beta_h$ was

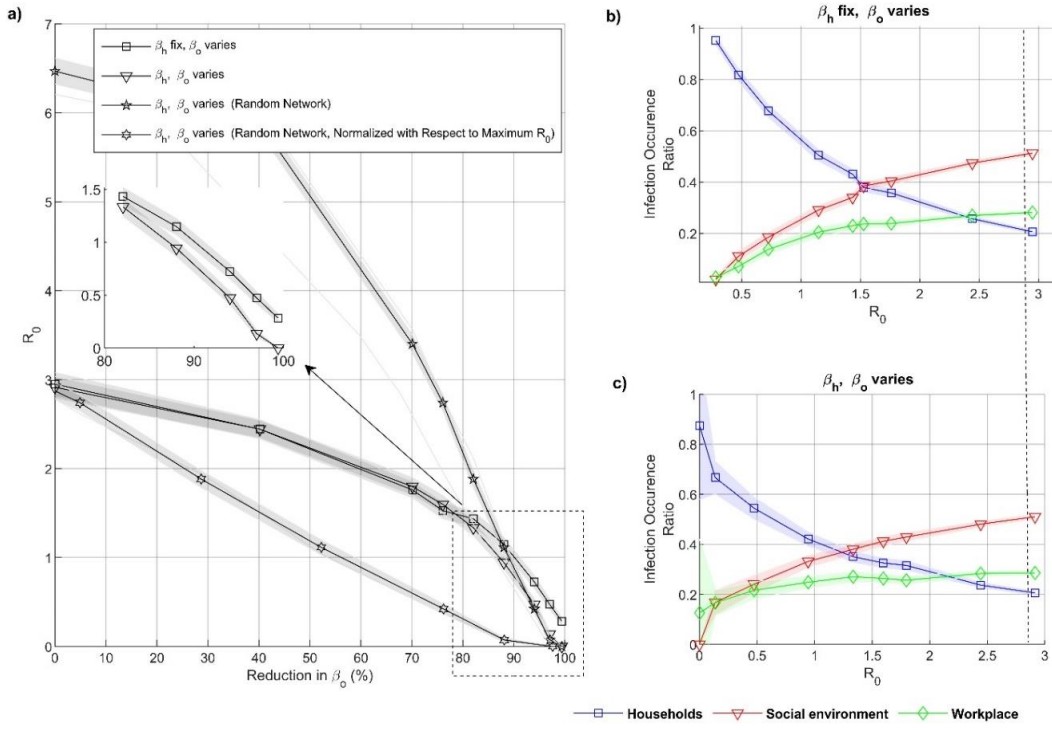

**Figure 2** Representation of effect of social distancing infection events. (A) Show the $R_e$ correspond for different reduction level in transmission probability. When agents applied social distance measure only in non-household (graph with square marks) and agents applied social distance measure in household and non-household (graph with triangular marks). The $x$-axis shows, the percentage of reduction of the transmission probability (see method). (B–C) Shows the infection occurrence ratio for $R_e$ in the household (blue graph), workplaces (green graph), and social environment (red graph), (B) when $\beta_h$ fix and $\beta_o$ varies, (C) when $\beta_h$ and $\beta_o$ varies. The dashed line demonstrates the COVID-19 $R_o$, the shaded areas in (A–C) give 95th confidence intervals.

kept constant, $R_e<1$. At $R_e<1$, the infections' environments do not change significantly. These results are further supporting the results in Fig. 2A. The decrease in household transmission rate only affects the dynamics at low $R_e$.

Additional simulations have been made with random networks. Two separate results are plotted: (1) the range of $\beta_o$ and $\beta_h$ are the same with the real-network, line with circles in Fig. 2A; (2) the maximum $R_e$ is taken as the 2.87 (same as the maximum of the real-network), line with diamonds in Fig. 2A.

For case 1, the random networks show much higher $R_e$. Except, when the transmission rate is reduced by more than 88%, the random network shows lower $R_e$ than the simulations with the real network ($\beta_h$ fixed), Figure 2A. Thus, the real network used is limited by the number of unique connections. The interactions are more clustered and local. On the other hand, the frequency of interactions among the selected agents is significantly higher.

$R_e$'s decreasing is much faster than the real networks (Fig. 2A). Thus, decreasing $R_e$ in real networks is harder and retains infection chains.

## Estimating contribution of household and non-household in stabilizing of COVID-19

To better understand the effect of household and non-household transmission on the spread of disease, we study the transmission chain of infections. In the earlier results, Fig. 2, we counted the place of infections in the transmission chain, namely, the household (H), social environment (S), and workplace (W). This can be regarded as the 0th order knowledge on the transmission chain—we first group S and W to non-household transmission (O) to get further information. We then study the third-order knowledge by counting three consecutive places of infections, *e.g.*, HOH (the first infection is in household, the infected person infected another person in O, and that person infected another person in H).

There are eight combinations of H and O: HHH, HHO, HOH, HOO, OHH, OHO, OOH, OOO. The combination OOO is dominant for Ro at 2.87, the default case Fig. 3. HHH is the least frequent. Thus, a decrease in $\beta_o$ affects the OOO-chain three times, whereas $\beta_h$ has a minor influence. Therefore, the spread of the disease (Ro at 2.87) on our real network depends much more sensitively on $\beta_o$ than $\beta_h$.

Similar analysis can be made for the real network when $R_e = 1$, $\beta_h$ is fixed or not. The combination OOO is significantly lower. The combinations with Hs are much more frequent. Therefore, the system becomes more sensitive to $\beta_h$.

Besides the real-world network, a random network was also simulated for $R_e$ at 2.87. For the correct comparison, the number of H and O edges is kept the same in random and real networks. The number of H connections is more than O connections. It is natural to assume that infections should occur more frequently on H-edges. Indeed, At $R_o = 2.87$, the frequencies of the combinations are the opposite for the random networks and the real network, Fig. 3. The combination of OOO dominates the real network, while the HHH is the highest random network. The discrepancy is that the unique pairs for H connections are scarce, but their frequency is significantly higher.

Overall, at high $\beta_h$ and $\beta_o$, both H and O infections are high, but the higher number of unique O pairs contributes more to the spread. In contrast, when the $\beta$ decrease, the infection at O diminishes faster than H because the high frequency of connections for the H-edges provides a higher infection probability.

## Calculating the influence of stay-at-home restriction during weekends in the spread of COVID-19

After investigating the effects of contagion probability, we have investigated the effect of stay-at-home restrictions on weekends. To understand the impact of stay-at-home restriction versus transmission probability, we have simulated three cases: free weekend without restrictions, restriction on Sunday, and restriction on Saturday and Sunday. We have simulated each restriction scenario when social distancing is applied in the workplace and social environment ($\beta_h$ fixed). The edge frequencies of the altered networks are given in Figs. 4A, 4B, and 4C. As detailed in the methods, the household edges replace the social environment edges to implement the weekend restrictions.

We have found that stay-at-home restrictions during the weekends cause a decrease in $R_e$. Figure 5A shows that $R_o$ drops from 2.95 to 2.71 for restrictions on Saturdays and to 2.56

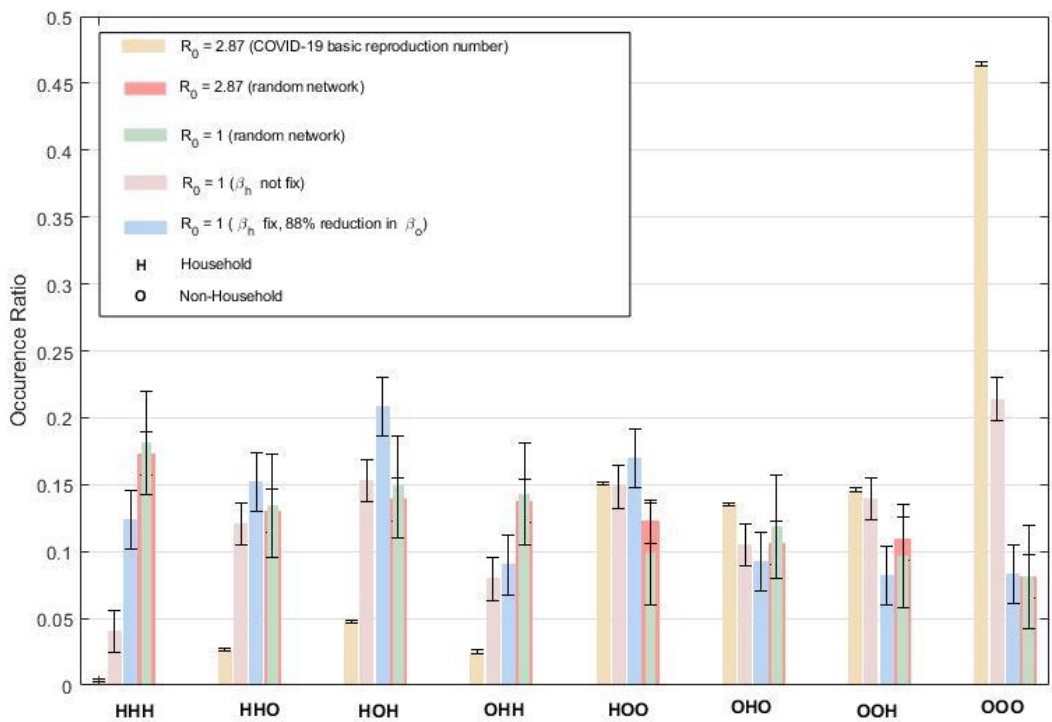

**Figure 3** **Third order transmission chain analysis.** Five cases were simulated. The simulations with random networks ($R_o = 2.87$, $R_o = 1$) are plotted overlappingly. For $R_o = 2.87$ the infections occurred the most as OOO combination for real network. Followed by combinations with two "O"s. Then tracked by single "O"s and lastly by HHH. There is a significant decrease in OOO combination for $R_o = 1$, while others are of comparable frequency. The random simulations are quite the opposite; the HHH combination has the most occurred ratio since total "H" edges are more than "O" edges.

when restrictions are on Saturdays and Sundays. Figures 5B, 5C and 5D show the ratio of infection occurrence ratio for increasing $R_e$ when implementing stay-at-home restrictions during weekends. When stay-at-home restrictions increase, the social environment's infection decreases, and work infections increase.

Expectedly, decreasing the social environment edges decreases infections in the social environment. However, the increase in household edges does not increase the number of household infections due to saturation. Overall, it leads to a decrease in $R_o$ (*Moreland et al., 2020*; *Czeisler et al., 2020*).

## Estimating impact of the daily working hours, stay-at-home restriction and transmission reduction level during weekends and weekdays on COVID-19

The subsequent alteration to the network is on working hours. We alter working hours by changing working edges. For example, if the working hours decreased from nine to eight hours, the work edges between 5 pm to 6 pm are converted to a sample from the edges between 6 pm to 11 pm, as detailed in the methods. The edge frequencies of the altered networks are given in supplementary information Figs. 6A, 6B, 6C, 6D, and 6E.

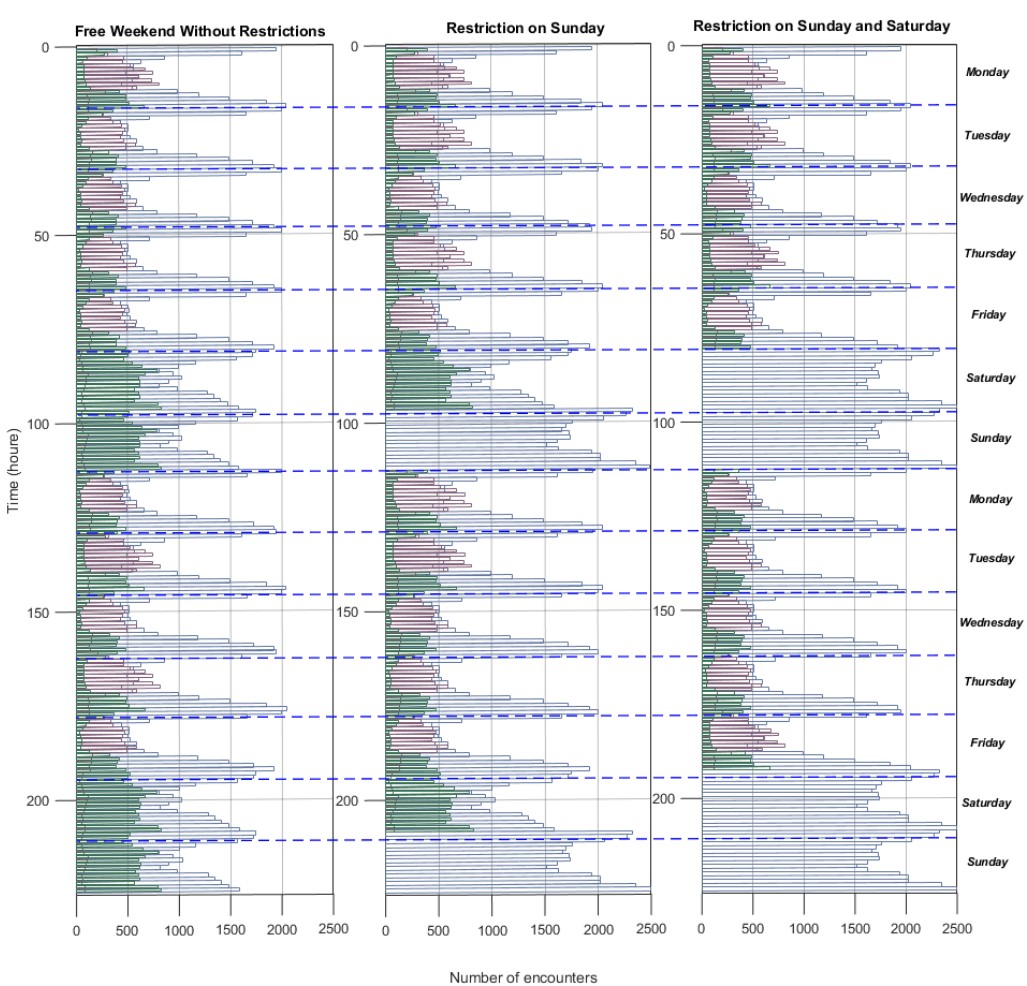

**Figure 4  The edge frequencies of the altered networks.** For (A) free weekend without restriction scenario, (B) restriction on Sunday, (C) restriction on Sunday and Saturday.

The last alteration is the vaccination. Different vaccination levels have been simulated. At the beginning of the simulations, a certain percentage of randomly selected individuals are vaccinated.

We simulate combinations of different vaccination levels, working hours, social distancing measures (transmission probability), and weekend restrictions: thirty thousand simulations were made. The number of parameters hinders us from understanding the interplay of parameters. One would like to understand the following functions $Re_{WT}$ (DW, SDM, SH, Vac), $Re_{delta}$ (DW, SDM, SH, Vac), where DW is the decrease in working hours, the SDM is the social distancing measure, the SH is the stay-at-home restriction (in weekends and in weekdays), Vac is the vaccination ratio.

Plotting for all simulations is hindered due to the multidimensionality of the results. Deducing a closed-form solution is also almost impossible. Instead, we wanted to approximate the solutions by fitting a multidimensional linear surface.

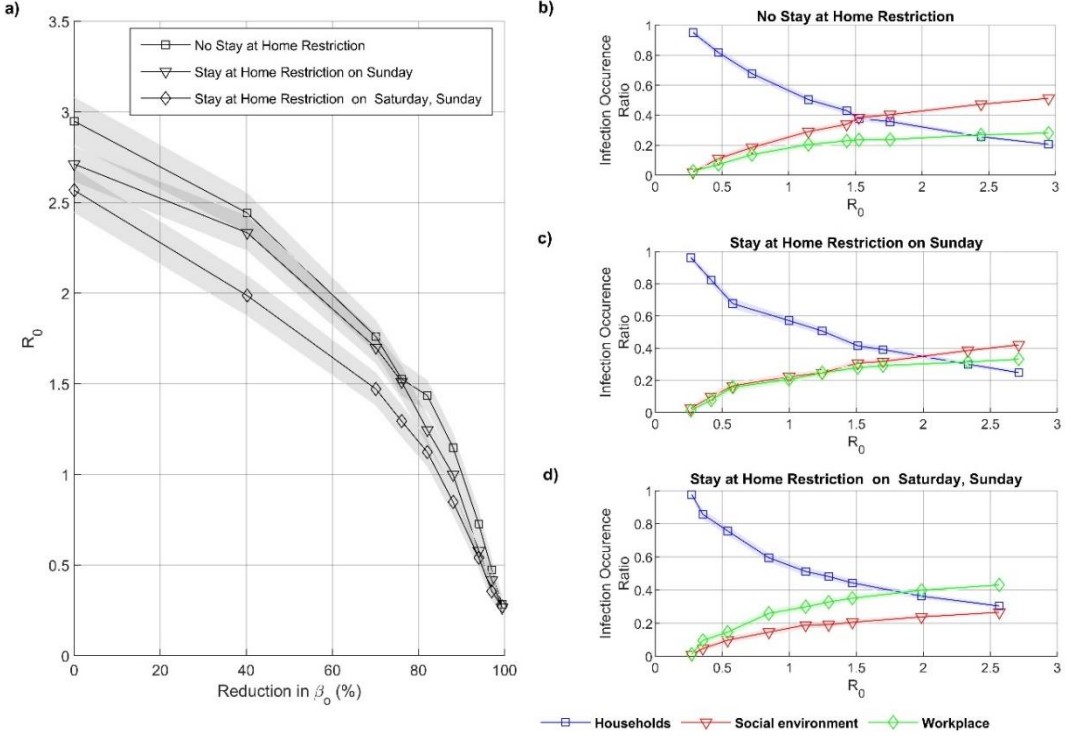

**Figure 5** **Representing the influence of stay-at-home restriction during the weekends, under social distance policy, in the spread of COVID-19.** (A) The basic reproduction number in three stay-at-home restriction scenarios: free weekend without restrictions (graph with square marks) restriction on Sunday (graph with triangular marks), restriction on Sunday and Saturday (graph with diamond marks). The x-axis demonstrates the percentage of reduction in transmission probability (see Method for more information). Here agents are only applied social distance measure in the non-household. (B–D) The infection occurrence ratio with respect to $R_o$ in the household (blue graph), workplaces (green graph), and social environment (red graph), for (B) 1st, (C) 2nd, and (D) 3rd stay-at-home restriction scenarios, respectively. The shaded areas in (A–D) give 95th confidence intervals.

$$Re_{WT} = \alpha_0 + \alpha_1 \cdot DW + \alpha_2 \cdot SH + \alpha_3 \cdot Vac + \alpha_4 \cdot SDM \tag{5}$$

$$Re_{delta} = \beta_0 + \beta_1 \cdot DW + \beta_2 \cdot SH + \beta_3 \cdot Vac + \beta_4 \cdot SDM. \tag{6}$$

The linear surfaces yielded a poor fit when we plotted against simulated data. Therefore, the mathematical expressions are made more complicated systematically. Firstly, the square term of SDM, Vac, SH, and DW are added to (5) (see Table S4 equ. B, C, D, and E) and Eq. (6) (see Table S5 equ. B, C, D, and E). Then, the root mean square errors (RMSE) of updated equations are calculated based on the simulated data and compared with the RMSE of Eqs. (5) and (6). Among these, the equation with the square of SDM reduces RMSE, but not sufficiently.

Secondly, the mathematical expressions are updated by adding a multiplication term (see Table S4 for WT and S5 for delta variant, equ. F, G, H, and j). Accordingly, the RMSE of the updated equation with multiplication term of the SDM parameter is less than the

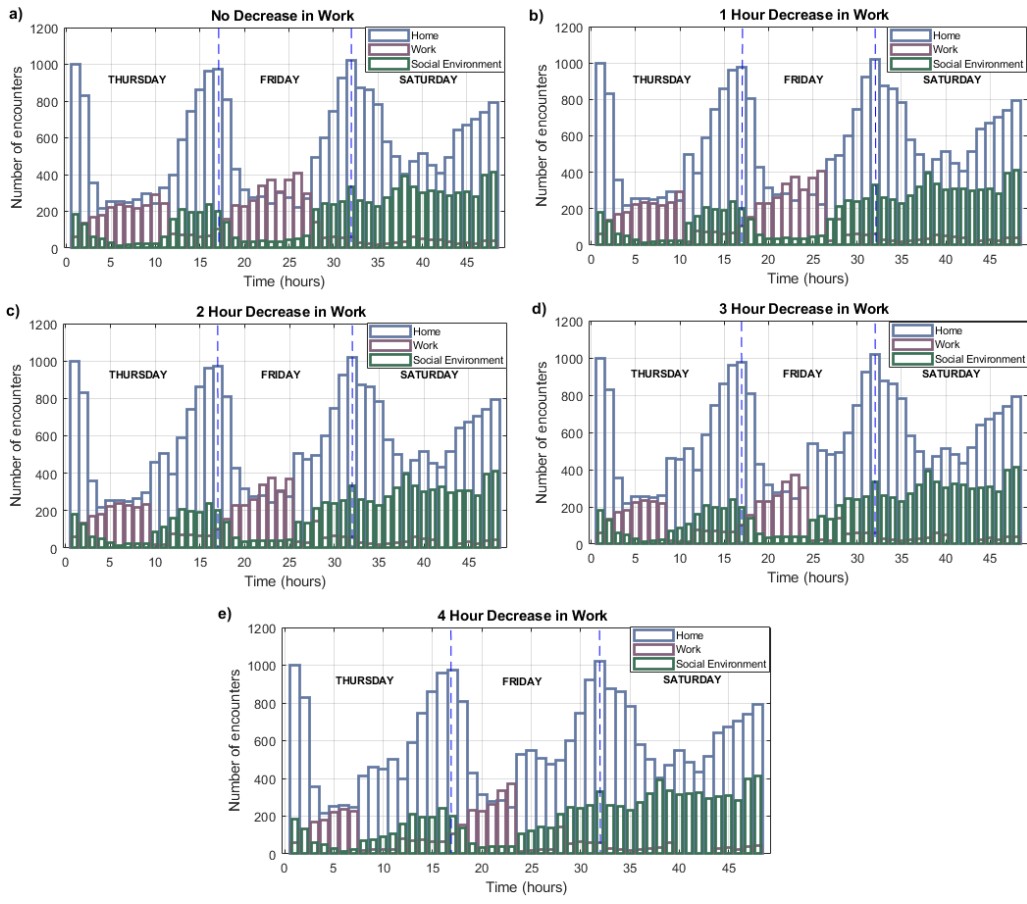

**Figure 6  The edge frequencies of the altered work hour.** For (A) no decrease, (B) 1 h decrease in work, (C) 2 h decrease in work, (D) 3 h decrease in work, (E) 4 h decrease in work.

equation with multiplication term of the Vac, DW, and SH. Additionally, it has a smaller RMSE than the equation with the square root of SDM. Consequently, it has been observed that the SDM is a sensitive parameter in Eqs. (5) and (6).

Thirdly, the mathematical expressions are updated by multiplying the four parameters (see Table S4 for the WT and S5 for delta variant, Equation K). However, it has approximately the same RMSE as the equation, which is updated by adding the multiplication term of the SDM parameters.

Next, the mathematical expressions are updated by adding a multiplication term with a square term (see Table S4 for WT and S5 for delta variant, equ. M, N, O, and P). Interestingly, the updated equation with multiplication and square terms of the SDM parameter has a smaller RMSE than the others. Therefore, it is predicted that updating the equations with the multiplication term and power term of the SDM parameter will reduce the RMSE. Considering this feature, finally, the equations are updated by adding the third power of the SDM parameter. Consequently, a substantial RMSE is obtained (see Table S4 for WT and S5 for delta variant, equ. S). The updated Eqs. (7) and (8) has a good fit

with the simulated reproduction number of WT COVID-19 (see Fig. S5) and delta variant COVID-19 (see Fig. S6), respectively. It is crucial to remember that our goal is to discover a trustworthy and simple expression that would allow us to examine the trade-off between our variables rather than the best fit.

$$Re_{WT} = \left( \alpha_0 + \alpha_1 \cdot DW + \alpha_2 \cdot SH + \alpha_3 \cdot \frac{Vac}{100} \right)$$
$$\cdot \left( \alpha_4 \cdot \frac{SDM}{100} + \alpha_5 \cdot \left( \frac{SDM}{100} \right)^2 + \alpha_6 \cdot \left( \frac{SDM}{100} \right)^3 \right) \tag{7}$$

$$Re_{delta} = \left( \beta_0 + \beta_1 \cdot DW + \beta_1 \cdot SH + \beta_2 \cdot \frac{Vac}{100} \right)$$
$$\cdot \left( \beta_3 \cdot \frac{SDM}{100} + \beta_4 \cdot \left( \frac{SDM}{100} \right)^2 + \beta_5 \cdot \left( \frac{SDM}{100} \right)^3 \right). \tag{8}$$

The expressions are valid for the range of the simulations, 0 to 4 h for the DW, 0 to 3 for the SH, 0 to 90 for the Vac (90 per cent vaccination), and 0 to 100 for the SDM. The scatter plots for the simulated and estimated values are given in Figs. 7A–7B. The coefficients of determination (R2) are 0.9722, and 0.9951 for WT and delta variant, respectively. After rearranging the Eqs. (7) and (8), the expressional relations are as follows:

$$Ro_{WT} = 2.76 \cdot (1 - 0.0125 \cdot DW - 0.072 \cdot SH - 0.54 \cdot Vac)$$
$$\cdot (1 - 0.56 \cdot SDM + 0.31 \cdot SDM^2 - 0.64 \cdot SDM^3) \tag{9}$$

$$Ro_{delta} = 3.44 \cdot (1 - 0.015 \cdot DW - 0.074 \cdot SH - 0.36 \cdot Vac)$$
$$\cdot (1 - 0.69 \cdot SDM + 1.21 \cdot SDM^2 - 1.41 \cdot SDM^3). \tag{10}$$

According to Eqs. (5) and (6), the trade-offs are easily measured. Changing working hours has a minor effect for both the WT and delta variants, while vaccination has the most effect. More specifically, for the WT, one day of the weekend restriction equals more than 4 h of work per weekday. In comparison, one day of restriction is equal to 13.6% vaccination. For the delta variant, the effect of the decrease in working hours and weekend restrictions act the same, while one day of restriction is equal to 20.2% per cent of vaccination. However, the relative effect of vaccination is 19.6% less effective than the WT.

It is important to note that the validity of functions is limited to the simulated ranges of parameters. Of course, full vaccination is expected to lower $R_e$ below 1. However, this is not the case for functions in Eqs. (5) and (6). This is expected because our simulated range is 0 to 90 per cent. After 90% vaccination Re depends non-linearly on Vac until reaching $R_e = 0$. The same discussion can be done for other parameters as well.

The SDM (transmission probability) varies from 0 to 100 per cent, unlike other parameters. As seen in Fig. 2A, the reduction in transmission probability reduces $R_e$ significantly after an 80% reduction.

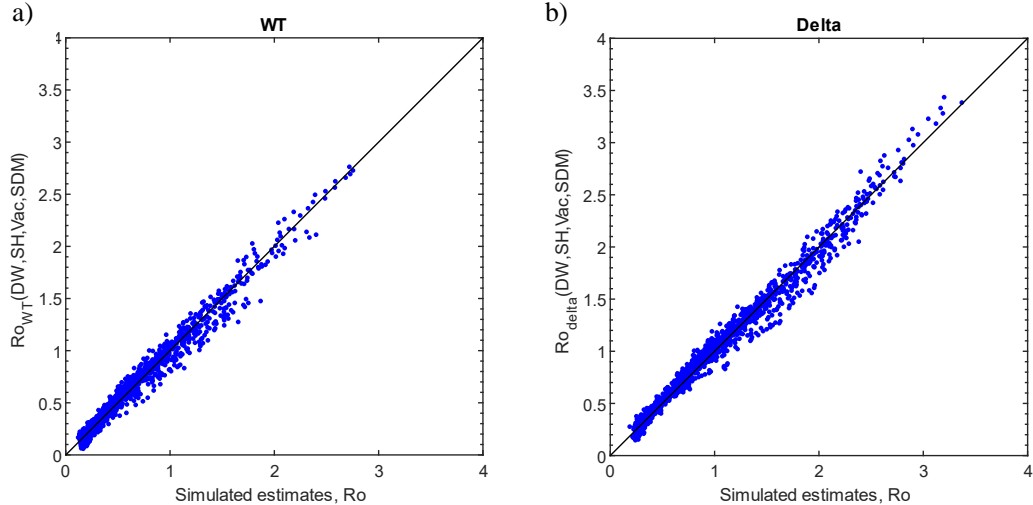

**Figure 7  Errors of the functions.** The estimations from the functions and the simulations have been plotted. (A) Wild type, (B) delta variant.

The parameters that we have simulated, DW and SH, vary marginally. For example, a decrease in work hours (DW) only changes from 0 to 4 h. In terms of working hours, the DW varies from 9 h to 5 h of work. In the simulated range, $R_e$ is not significantly affected. Perhaps simulating for the remaining hours, from 9 to 0 h of work, would show the non-linearity that we have suspected. However, the point we want to make is that a marginal decrease in working hours (several hours) leads to a minimal decrease in $R_e$. The working hours must be decreased substantially to decrease $R_e$ significantly. However, its burden on economic activity would be profound.

The range for SH (stay-at-home restrictions) has been simulated from 0 to 4 days. The meaningful range is 0 to 2 days, the weekend. Two days of restrictions are equal to 26.8% vaccination for the WT, 41% for the delta type. However, it should be noted that the simulations on the delta variant with 90% vaccination and weekend restrictions have about $R_e = 2$ as the delta variant starts with a higher $R_e$. The weekend restriction on the wild type is more meaningful. The simulations with 90% vaccination with weekend restrictions have $R_e = 0.98$.

## DISCUSSION

From Fig. 2, we speculate that the household transmission reaches its capacity at a low transmission rate inside the household ($\beta_h$) due to abundant link frequency. The large $\beta_h$ cannot lead to an additional increase in $R_e$ value in the population. Therefore, the increase in household transmission contributes to the resilience of the eradication but does not contribute significantly to the overall spread. *Bulfone et al. (2020b)* found that indoor transmission probability was very high compared to outdoors. This result confirms our finding by demonstrating that people are likelier to become infected inside the household. More interestingly, 1,587 close contacts of confirmed cases with COVID-19

have been traced in a study. Among them, 88.1% (1398) are their family members. Furthermore, *Shen et al. (2020b)* found that contact within households is responsible for roughly 70% of SARS-CoV-2 transmission when widespread community control measures are in place. This finding is in line with our results because we showed that a high contact rate is responsible for transmission inside the household. When there are harsh measures implemented outside, most people stay in their homes. Due to frequent contact inside the household, the transmission trend is continuous. Consequently, it prevents the eradication of the disease. In line with our predictions. *Nande et al. (2021)* have found that household transmission is a severe factor for the eradication of an epidemic. The involvement of household transmission against the eradication is a known fact. Our other half of the claim, which is that the increase in household transmission rate does not contribute to the dissemination of the disease, should be considered very carefully. We argue that even low in-house transmission rates saturate the impact on overall reproduction number; the effect is flooded due to abundant contact frequency. The household transmission rate does not limit the reproduction number, a low rate is enough, but the outside transmission rate limits it.

The way we investigate the impact of human behavior on the household, workplace, and social environment can be generalized to other infectious disease epidemics. MERS (*Liu et al., 2020*) is an infectious disease that first appeared in Saudi Arabia in September 2012. Basic reproduction number ($R_o$) for the MERS has ranged from 0.42 to 0.92 (*Breban, Riou & Fontanet, 2013*; *Fisman, Leung & Lipsitch (2014)*; *Cauchemez et al., 2014*; *WHO, 2021*). In 2017, a cluster of the MERS was reported from the Al-Jawf region, Saudi Arabia, including seven cases, six of which were household contacts (*WHO, 2021*). According to our results (Fig. 2A), since $R_e<1$ for the MERS, most infections have occurred in households, which is in line with earlier work. It may also be noted that household infections can increase the resilience of eradication but do not significantly contribute to the spread.

Understanding the tradeoff between non-pharmaceutical interventions and vaccination is essential for controlling, policy making, and managing of COVID-19 pandemic and economic activities. In literature, numerous valuable theoretical deductions were made on simpler models (*Estadilla et al., 2021*; *Gumel et al., 2021*; *Iboi, Ngonghala & Gumel, 2020*; *Rao & Brandeau, 2021*). Also, valuable numerical deductions were made on complex models (*Moore et al., 2021a*; *Goldenbogen et al., 2022*). *Betti et al. (2021)* developed a modified SIR compartmental model. They compared vaccination and non-pharmaceutical intervention by linking the effective basic reproduction number with non-pharmaceutical intervention and vaccination percentage. They showed that both non-pharmaceutical and vaccination intervention should be considered to control the pandemic. Their finding is similar to ours because we have also demonstrated that it is important to consider all measures. Our contribution is that we have performed complicated and as real-life simulations on the real network as possible to reach at our results. This network is also inherently heterogeneous. Understanding the numerical solutions is troublesome when there are many parameters. We have guessed a simple mathematical expression for the whole simulation to circumvent that problem. Indeed, the results can be approximated with a simple form. The simple functions allowed us to compare between vaccination

and non-pharmaceutical intervention such as decrease in working hours, weekend restrictions, and decrease in transmission probability (social distancing measures) by linking the effective basic reproduction number with non-pharmaceutical intervention and vaccination percentage. In addition, any measure that is not taken to almost completion does not significantly affect the outbreak. This means that at least one measure must be performed to its maximum level. Furthermore, mixing marginal effects would not be strong enough so that Re = 1. For example, the required reduction in transmission probability for $R_e = 1$ is 84.6%. With weekend restrictions, it only decreases to 80.1%. Worse off, when the work hours are deducted by 3 h the percentage only decreases to 83.6%.

Our results show that vaccination and transmission reduction are almost interchangeable. *Andersson et al. (2021)* found that vaccination reduces the need for social distancing. Since, in our model, we proxy the social distance by transmission reduction; hence, our finding is compatible with their finding. In our simulations, 90% vaccination had not brought Re below 1 (when no other measure was implemented), for both wild type and delta variants. Many countries, such as Singapore and the United Arab Emirates, have more than 80% (*Reuters, 2021*) vaccination ratios, but they had cases when the delta variant was dominated (*Worldometer, 2021a*; *Worldometer, 2021b*). *Voigt, Omholt & Almaas (2022)* have investigated the impact of vaccination on reproduction number. They have found that for both delta and wild-type variants vaccinating 80% of population does not bring reproduction number below 1.

There are some areas for improvement in our work. The children under thirteen were not included in this data (*Goeyvaerts et al., 2018*). Children play a crucial role in bringing infection into the household (*Jing et al., 2020*). Since there are only three days of data, we reuse the data five times. We tried to decrease the effect of this by only performing brief simulations. Simulations last for 14 days, and we only allow for a maximum of around twenty infections per simulation. We recommend for social network data miners increase their sample size during data collecting from a population. It is important to note that our results depend heavily on the real-world network. We claim that our results are correct on our real-world network.

## CONCLUSIONS

This article rigorously explores the impact of human behaviour, economic activity, vaccination and social distancing in the context of the control and containment of COVID-19. In this respect, an agent based model based on a time-dynamic real network with stochastic transmission events has been created. The network has been successfully categorized as household, workplace, and social environment.

The conditions needed to mitigate the spread of wild-type COVID-19 and the delta variant have been analyzed. By our agent-based model, simulations that are complicated and in agreement with real life as much as possible have been performed on the real network. The results of simulations have been interpolated with a simple form. Consequently, the interaction between pharmaceutical and non-pharmaceutical intervention in the containment of COVID-19 has been investigated.

Our simulation results showed that the effect of stay-at-home restriction, decreased working hours, social distancing and vaccination are different. It has been seen that vaccination and social distancing can practically replace each other. Our findings highlight the ineffectiveness of imposing stay-at-home restrictions and reducing working hours without vaccination. It has been conducted that the least effective is the reduction of working hours. Teaching people how to lower their transmission probability by social distancing and finding effective vaccines surpasses the effect of the stay-at-home restrictions and working hour reduction. The current findings suggest that policymakers, under pandemic conditions, reduce work-related activities as a last resort and should probably not do so when the impacts are minimal. The economic ramifications clearly, as seen in 2022, outweigh the minimal effect we would achieve.

### Funding

This work was supported by TUBITAK, 2232 - International Fellowship for Outstanding Researchers, Project number 118C244. The funders had no role in study design, data collection and analysis, decision to publish, or preparation of the manuscript.

### Grant Disclosures

The following grant information was disclosed by the authors:
TUBITAK, 2232 - International Fellowship for Outstanding Researchers: 118C244.

### Competing Interests

The authors declare there are no competing interests.

### Author Contributions

- Ramin Nashebi conceived and designed the experiments, performed the experiments, analyzed the data, prepared figures and/or tables, authored or reviewed drafts of the article, and approved the final draft.
- Murat Sari analyzed the data, authored or reviewed drafts of the article, and approved the final draft.
- Seyfullah Kotil conceived and designed the experiments, performed the experiments, analyzed the data, prepared figures and/or tables, authored or reviewed drafts of the article, and approved the final draft.

### Data Availability

All codes are available at GitHub: https://github.com/raminnashebi/MATLAB-Codes.git; Ramin Nashebi, Murat Sari, & Seyfullah Kotil. (2022). Using a real-world network to model the trade-off between stay-at-home restriction, vaccination, social distancing and working hours on COVID-19 dynamics [Data set]. Zenodo. https://doi.org/10.5281/zenodo.7250128.

## Supplemental Information

Supplemental information for this article can be found online at http://dx.doi.org/10.7717/peerj.14353#supplemental-information.

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

## FURTHER READING

**Firth JA, Hellewell J, Klepac P, Kissler S, Kucharski AJ, Spurgin LG. CMMID COVID–19 working group. 2020.** Using a real-world network to model localized COVID-19 control strategies. *Nature Medicine* **26**:1616–1622 DOI 10.1038/s41591-020-1036-8.

**Galanis G, Di Guilmi C, Bennett DL, Baskozos G. 2021.** The effectiveness of Non-pharmaceutical interventions in reducing the COVID-19 contagion in the UK, an observational and modelling study. *PLOS ONE* **16(11)**:e0260364 DOI 10.1371/journal.pone.0260364.

**Groendyke C, Combs A. 2021.** Modifying the network-based stochastic SEIR model to account for quarantine: an application to COVID-19. *Epidemiologic Methods* **10(s1)**:20200030 DOI 10.1515/em-2020-0030.

**Karaivanov A. 2020.** A social network model of COVID-19. *PLOS ONE* **15(10)**:e0240878 DOI 10.1371/journal.pone.0240878.

**Mahendra P, Shailendra S, Michael W, Iryna Z. 2021.** Optimal governance and implementation of vaccination programmes to contain the COVID-19 pandemic. *Royal Society Open Science* **8(6)**:210429 DOI 10.1098/rsos.210429.

**Patel MD, Rosenstrom E, Ivy JS, Mayorga ME, Keskinocak P, Boyce RM, Hassmiller Lich K, Smith RL, Johnson KT, Swann JL. 2021.** The joint impact of COVID-19 vaccination and non-pharmaceutical interventions on infections, hospitalizations, and mortality: an agent based simulation. *Preprint. medRxiv* DOI 10.1101/2020.12.30.20248888.

**Sewell DK, Miller A. 2020.** Simulation-free estimation of an individual-based SEIR model for evaluating nonpharmaceutical interventions with an application to COVID-19 in the district of Columbia. *PLOS ONE* **15(11)**:e0241949 DOI 10.1371/journal.pone.0241949.

**United States of America Department of State. 2021.** COVID-19 Information. *Available at* https://tr.usembassy.gov/covid-19-information-2/.

**Volz E. 2008.** SIR dynamics in random networks with heterogeneous connectivity. *Journal of Mathematical Biology* **56**:293–310 DOI 10.1007/s00285-007-0116-4.

**Yu H-J, Hu Y-F, Liu X-X, Yao X-Q, Wang Q-F, Liu L-P, Yang D, Li D-J, Wang P-G, He Q-Q. 2020.** Household infection: the predominant risk factor for close contacts of patients with COVID-19. *Travel Medicine and Infectious Disease* **36**:101809 DOI 10.1016/j.tmaid.2020.101809.