# Peer review of "Using a real-world network to model the trade-off between stay-at-home restriction, vaccination, social distancing and working hours on COVID-19 dynamics"

_PeerJ, doi:10.7717/peerj.14353_

## Round 0.1 · original submission · Major Revisions

Your manuscript has been reviewed and assessed by three reviewers, and all of them agree with the fact that there are still a few points that need to be addressed. The comments of the reviewers are included at the bottom of this letter. Reviewers indicated that methods, discussion, and conclusion sections should be improved. We would be glad to consider a substantial revision of your work, where the reviewer’s comments will be carefully addressed one by one.

Reviewer 1 ·

Basic reporting

The review is of broad and cross-disciplinary interest. The topic and content discussed in this manuscript are within the scope of the journal.

Experimental design

Experimental design
Comments: The survey methodology is consistent with comprehensive and unbiased coverage of the subject.
The organization and subsections are also appropriate. The manuscript is structured and presented in a reader-friendly manner.

Validity of the findings

The authors developed an agent-based model based on a time-dynamic graph with stochastic transmission events. For the WT, it has been found that a 13% increase in vaccination impacts the reproduction number. The author observed that the change in household transmission rate does not significantly alter the Re. Household infections are not limited by transmission rate due to the high frequency of connections. For COVID-19’s specifications, the Re depends on the non-household transmissions rate.

Comments;
1) Why did the authors choose Friday, Saturday, and Sunday for research?
2) In the Discussion and Conclusion sections, authors should interpret the results, place them in the context of previous findings, and explain what they mean for possible real-life applications.
3) In the results section, please explain Figure 1a.

Reviewer 2 ·

Basic reporting

• A copy edit is required. There are grammar errors and awkward writing construction.
• Presentation needs to be clarified, as does the narrative. The abstract, as an example, is very unclear and needs to be rethought, particularly the conclusion.

Experimental design

There is some imprecision with the description of the methods, and missing information. I have also flagged in the attachment. While the research question is clear, what this paper is contributing to the existing literature landscape on COVID-19 mitigation outcomes modeling is not well articulated and should be done up front. Some of the justifications provided for modeling decisions and assumptions do not make sense and need to be backed up with literature and/or clarified.

Validity of the findings

As noted above, there is information missing in the Methods. The results section contains both Methods and Discussion and needs to be rewritten. Otherwise the section by section walk through is clear.
Conclusions are not well stated and need to be clarified.

Annotated reviews are not available for download in order to protect the identity of reviewers who chose to remain anonymous.

Reviewer 3 ·

Basic reporting

The study aims to report on the effects of different pharmeceutical and non-pharmaceutical measures on the spread of the infectious disease such as COVID19. The study uses 3-days of data and simulates for 14 days as initialisation phase and then simulate for more realistic outcome. While the study is novel and has merits the reporting could be improved.
- The report includes repetitive sentences: i.e. lines 276-280 are the same as lines 298-302.
- Some sentences are not defined clearly: i.e.
line 68 "These countries recommend stay-at-home orders only during weekends." there is no explanation in what period and for what cause?
line 115: "The simulations that were made depend on four parameters", there is no mention of what parameters.
Are lines 320 & 322 required?
line 517: "our simulations show that changing the working hours or weekend restrictions will only make people more frustrated without vaccination or teaching people how to lower their transmission probability significantly." The sentence is mis-leading, as I doubt if the results of simulations done by authors can illustrate people's frustrations.
- "Vaccination and transmission reduction are almost interchangeable." is repeated in few occasions.

Experimental design

The methodology used is ambiguous in some parts:
- line 211: "We run five hundred trial simulations for each scenario. " Please clarify what scenarios? There is also mention of first and second scenario in line 316 with no clarification.
- In equations 1 and 2: what is d_ij? Is it distance (shown as dis_ij in the same equations)?
- How were the equations 7 and 8 obtained?
- Line 309: "The agents that are vaccinated are chosen randomly", What percentage of population under study was chosen to be vaccinated during the 14-day period of study?

Validity of the findings

no comment

Additional comments

Figures could be improved in terms of readability

---

## Round 0.2 · accepted · Accept

The authors addressed the reviewers' concerns and substantially improved the content of the manuscript. So, based on my own assessment as an academic editor, no further revisions are required and the manuscript can be accepted in its current form.

Reviewer 1 ·

Basic reporting

N/A

Experimental design

N/A

Validity of the findings

N/A

Additional comments

N/A

Reviewer 3 ·

Basic reporting

The revised manuscript has been improved tremendously in terms of clarity and flow of the concepts defined.

Experimental design

The experiments have been designed and revised appropriately.

Validity of the findings

The findings are novel and explained appropriately

Additional comments

The manuscript is revised appropriately and all comments are responded with details. some experiments have ben repeated/revised to improve the quality of the paper.